# Fabrication and Hypersonic Wind Tunnel Validation of a MEMS Skin Friction Sensor Based on Visual Alignment Technology

**DOI:** 10.3390/s19173803

**Published:** 2019-09-03

**Authors:** Xiong Wang, Nantian Wang, Xiaobin Xu, Tao Zhu, Yang Gao

**Affiliations:** 1Hypervelocity Aerodynamics Institute, China Aerodynamics Research and Development Center (CARDC), Mianyang 621000, China; 2College of Information Engineering, Southwest University of Science and Technology, Mianyang 621010, China

**Keywords:** skin friction measurement, hypersonic wind tunnel, MEMS skin friction sensor, visual aligning, validation

## Abstract

MEMS-based skin friction sensors are used to measure and validate skin friction and its distribution, and their advantages of small volume, high reliability, and low cost make them very important for vehicle design. Aiming at addressing the accuracy problem of skin friction measurements induced by existing errors of sensor fabrication and assembly, a novel fabrication technology based on visual alignment is presented. Sensor optimization, precise fabrication of key parts, micro-assembly based on visual alignment, prototype fabrication, static calibration and validation in a hypersonic wind tunnel are implemented. The fabrication and assembly precision of the sensor prototypes achieve the desired effect. The results indicate that the sensor prototypes have the characteristics of fast response, good stability and zero-return; the measurement ranges are 0–100 Pa, the resolution is 0.1 Pa, the repeatability accuracy and linearity are better than 1%, the repeatability accuracy in laminar flow conditions is better than 2% and it is almost 3% in turbulent flow conditions. The deviations between the measured skin friction coefficients and numerical solutions are almost 10% under turbulent flow conditions; whereas the deviations between the measured skin friction coefficients and the analytical values are large (even more than 100%) under laminar flow conditions. The error resources of direct skin friction measurement and their influence rules are systematically analyzed.

## 1. Introduction

Skin friction is an important physical quantity in aerodynamics, and especially in hypersonic flow fields. Direct skin friction measurements in hypersonic wind tunnels mainly depend on various conventional skin friction balances [1,2,3,4,5]. However, with their constraints of low sensitivity, large volume, temperature effects and other defects, skin friction balances cannot be widely applied. MEMS skin friction sensors have high sensitivity and stability, small volume, low cost and other positive traits, thus, they can be potentially used to measure skin friction and its distribution in hypersonic wind tunnels. These are very important for vehicle design.

There are many studies on MEMS skin friction sensors, including various comb capacitance types and piezoresistive types, which are mainly used for skin friction measurements in low wind tunnels. A MEMS skin friction sensor designed by Jiang et al. in 2001 [6], adopted a supporting cantilever and plate differential capacitance sensing with a measurement range of 0.1–2.0 Pa, and was used in a low wind tunnel. Another MEMS skin friction sensor designed by Meloy et al. in 2011 [7], adopted four supporting cantilevers and comb capacitance sensing and had a measuring range of 0.1–5.0 Pa, however its floating element and comb capacitances were exposed in the flow field, thus it could be only used in low wind tunnels with pure gases. Direct skin friction measurements under hypersonic conditions are strongly required now, however in hypersonic wind tunnels, they are mainly implemented by using conventional skin friction balances. A strain gauge balance developed by Schetz in 2010 [5], was used to measure skin friction in a hypersonic wind tunnel at *M* = 4 with low sensitivity, poor temperature stability and large volume, thus it couldn’t be used to accurately measure skin friction and its distribution for vehicles. Besides, the larger normal loads in hypersonic flow fields present many difficulties to design and fabricate feasible MEMS skin friction sensors.

A novel MEMS skin friction sensor was developed by us [8,9], which adopted the principle of a floating element even with the measured wall and the signal output microstructure being isolated from the hypersonic flow field. Its test-head was connected to a clamped-clamped beam with sensing capacitances through a supporting pole. Thus, skin friction sensed by the test-head was transferred to the clamped-clamped beam and drove bilateral capacitance sensing vibratory electrodes to generate a torsional angle displacement. Then, the measured skin friction could be solved by differentially calculating the variations of the sensing capacitances. The results of static calibration and validation in a hypersonic wind tunnel indicated that the sensor prototypes presented high sensitivity, stability and anti-jamming, the sensing structure and encapsulation format were suitable for skin friction measurement in hypersonic flow fields [9], however, limited by assembly and packaging fabrication errors, its test accuracy couldn’t be estimated.

Aiming at addressing the demerit of the test accuracy induced by fabrication errors, a novel fabrication scheme based on visual alignment is proposed, which mainly involves the optimization of the sensor-head and interface circuit, precise fabrication, micro-assembly based on visual alignment, prototype fabrication, static calibration, validation and exploration of its dynamic calibration in a hypersonic wind tunnel.

## 2. Sensor Optimization

The sensor-head of our MEMS skin friction sensor is optimized with one pair of differential capacitances, and its interface circuit is accordingly optimized with one set of AD7747 (Analog Devices Inc., Norwood, MA, USA) sensing components to minimize its encapsulation volume and reduce existing fabrication errors. The optimized overall scheme and sensor-head structure are shown in Figure 1. The MEMS skin friction sensor mainly contains the sensor-head, interface circuit and package shell (including pedestal and cover). Its sensor-head mainly contains a floating element (including the test-head and supporting pole), a silicon microstructure (including frame, clamped-clamped beam and vibratory electrodes) and a glass base (including fixed metal electrodes). The upper surface of test-head is its sensing surface, and will be installed evenly with the measurement wall.

### 2.1. Sensor-Head Optimization

Combining material mechanics and electrostatics theories, the structure of the sensor-head and its parameters are optimized. The angle displacement *θ* of vibratory electrodes induced by skin friction moment is given by [10]:(1)$$\theta =\frac{T_{s}}{K}=\frac{1+\upsilon }{\beta }\frac{l_{1}h_{3}}{Ew^{3}h}\tau_{w}A,$$

According to conformal transformation theory [11], the differential sensing capacitance Δ*C* of unparallel rectangle electrodes with the torsional angle *θ* between the glass base and the silicon microstructure is:(2)$$\Delta C\approx \frac{\varepsilon_{0}\left(2w_{1}+w_{2}\right)w_{2}w_{3}}{h_{0}{}^{2}}\frac{1+\upsilon }{\beta }\frac{l_{1}h_{3}}{Ew^{3}h}\tau_{w}A,$$

It can be seen from Equation (2) that there exists good linearity between the differential capacitance output Δ*C* and the measured skin friction *τ_w_*.

The measurement range of the MEMS skin friction sensor is designed as 0–100 Pa, and the relevant differential capacitance output Δ*C* is 0–2 pF. The resolution is 0.05 Pa/Ff, and the initial value of each sensing capacitance is 9 pF. Combined with the designed sensor indexes and the sensitivity model, the sensor-head is optimized, and its main structural parameters are achieved, as shown in Table 1.

### 2.2. Interface Circuit Optimization

Because the sensor-head is optimized as one pair of differential capacitances, its interface circuit is accordingly optimized with one set of AD7747 sensing components [9], as shown in Figure 2.

The interface circuit not only needs to detect the weak differential capacitances of the MEMS skin friction sensor, but also needs to consider the high precision connections between the sensor-head and package shell. Printed Circuit Board (PCB) technology for interface circuits can’t satisfy the demands of sensor micro-assembly with high precision [9], thus precise ceramic-based micro-strip circuit technologies are adopted to implement the miniature interface circuit with high precision. The layout of the sensor-head, AD7747, Single Chip Micyoco (SCM), fixed resistors and capacitances for a ceramic-based interface circuit is optimized.

## 3. Sensor Fabrication Based on Visual Alignment

The MEMS skin friction sensor can be divided into five key parts, including a silicon microstructure with single-crystal silicon material, a glass base with Pyrex glass material, an interface circuit with ceramic material, and a floating element and package shell made with aluminum alloy material. These key parts will be fabricated separately, and then assembled. The silicon microstructure and glass base will be bonded as a silicon-glass microstructure to output sensing signals, where the MEMS technologies are adopted. Micro-strip circuit technology is adopted for the interface circuit, and precision finishing technologies are adopted for the floating element and package shell.

### 3.1. Silicon-Glass Microstructure Fabrication

Silicon-glass microstructures are fabricated via MEMS technologies, and they mainly involve photolithography with double-sided alignment and deep reactive ion etching (DRIE) for the silicon microstructure, chromium-gold metal deposition and electrode fabrication for the glass base, silicon-glass anode bonding and wheel scribing, and other MEMS technologies. Silicon microstructures are fabricated from a single-crystal silicon wafer of 4 inches in diameter and 400 μm thickness, and mainly involves oxidation, photolithography and DRIE, as shown in [9].

The MEMS process technologies of the glass base mainly involve metal deposition and electrode etching, and the MEMS processes are shown in Figure 3. A Corning Pyrex 7740 glass wafer with 4 inches diameter and 500 μm thickness from Valley Design Corporation (Valley Design Corp, Shirley, MA, USA) is adopted to fabricate the glass base, whose thermal expansion coefficient is close to that of single crystal silicon material. Firstly, composite metal films with 200 nm chromium film and 1 μm gold film are prepared via vapor deposition. Then, the layout of the metal electrodes is fabricated via photolithography. Finally, the metal electrodes are etched by chromium-gold etching solutions. 

The fabricated silicon microstructures and glass bases are aligned via double-sided alignment technology and the alignment masks fabricated in advance, and they are bonded together via anode bonding technology to keep their rigid connection. The bonded microstructures are separated by the wheel scribing method to maintain their 10 µm geometrical precision. The initial capacitances and dimensions of the separated silicon-glass microstructure are tested and selected, as shown in Figure 4.

### 3.2. Ceramic-Based Interface Circuit Fabrication

The ceramic-based interface circuit adopts precise micro-strip circuit technologies, which mainly involve substrate cutting, double side metal spattering, plating metal layer thickening, photolithography and etching, as shown in Figure 5a. Firstly, the ceramic substrates are cut, polished and precisely ground with 50 µm geometrical precision, and they are drilled via ultrasonic drilling technology. Then, the metal films are fabricated via double side metal spattering and plating metal layer thickening technologies, the wires and the solder pads are fabricated via photolithography and etching. Finally, all electronic components are welded on the double sides of the ceramic substrates. The prototype of the ceramic-based interface circuit is shown in Figure 5b.

### 3.3. Floating Element and Package Shell Fabrication

Limited by the fabrication errors of floating elements and package shells and the manual assembly process, the sensor prototypes showed poor assembly and package precision; thus, the accuracy of skin friction measurements couldn’t be estimated [9]. As a result, floating elements and package shells are fabricated with aluminum alloy material via precise machining technologies. The package shells are fabricated via precision turning and milling technologies with 10 µm geometrical precision. The floating element is a special-shaped bar with a larger slenderness ratio, and it is fabricated via precision meter lathe and uses a roller tooling fixture to keep its geometrical precision. The sample parts of the floating elements and package shells are shown in Figure 6.

### 3.4. Sensor Micro-Assembly Based on Visual Alignment

Studies show that the accuracy of skin friction measurements is very closely related to the system errors of sensors and the flow characteristics of the measured wall [12]. However, the sensor-head adopts a mesoscopic quasi-tridimensional structure and the fabrication scheme of the sensor is divided into several key parts, which are fabricated separately, and then assembled. Thus, the circular test-head of the floating element must possess high concentricity with the circular hole in the upper face of package cover, and also high levelness with the upper face of package cover and the measured wall, excepting intrusive or recessive areas. Accordingly, special micro-assembly technologies with high precision must be studied for this sensor.

These key parts mainly include a silicon-glass microstructure, ceramic-based interface circuit, floating element, package pedestal and package cover. Aiming at meeting the requirements of assembly precision and the structural features of key parts, a novel micro-assembly method based on visual alignment and micro-operating assembly is presented. It mainly involves five procedures, including silicon-glass microstructure and interface circuit assembly, floating element and silicon-glass microstructure assembly, sensor-head and package pedestal assembly, wiring, and final encapsulation. The micro-assembly technological procedures are shown Figure 7, and the detailed micro-assembly processes are presented as follows:The locating flexures are adopted to implement preliminary alignment with 100 μm precision for silicon-glass microstructure and ceramic-based interface circuit, and then the two parts are fixed via an air-cured epoxy resin.The circular hole in the geometric center of silicon microstructure and the locating axis of floating element are used to implement their alignment with 10 μm precision, and then the two parts are fixed via an air-cured epoxy resin. As a result, the sensor-head assembly is finished.The package pedestal is placed in the locating flexure, and the package cover is installed. Then, the circular hole in the upper surface of the package cover is recorded by the machine vision camera, and its center position is identified via a canny edge detection method with ±5 μm precision [13].Move away the package cover, suck and place the sensor-head in the locating flexure. Record and identify the center position of its test-head via the camera and canny edge detection, and calculate the deviation of the center position between the circular hole in the package cover and the test-head. Then, carry and drive the sensor-head to match the center position of the circular hole in the package cover with ±20 μm precision.Lead the wires between the pads of the sensor-head and the interface circuit via a spot-welder. Install and fix the package cover, then the sensor micro-assembly are finished. Sensor prototypes are shown in Figure 8a.

The gaps between the floating element and the package cover of sensor prototypes based on visual alignment are almost 100 ± 30 μm, the gaps of the sensor prototypes assembled by hand are almost 300–500 μm, and the comparison of the package effect for the two-stage sensor prototypes is also shown in Figure 8b,c. The precision of fabrication and micro-assembly for sensor prototypes based on visual alignment achieves the desired packaging effect.

## 4. Static Calibrations

Because the volume of the MEMS skin friction sensor and the measured skin friction are very small, the conventional hanging weight calibration method used for skin friction balances is no longer feasible [3,4]. Thus, the centrifugal force equivalent method and the single-spindle rotary loading platform are adopted to statically calibrate the sensor prototypes. The static calibration device for sensor prototypes is shown in Figure 9.

Given the structural parameters and the mass of the sensor-head and the distance between the fixing position of the sensor prototypes and the rotation center of the rotary loading platform, the centrifugal force equivalent to skin friction sensed by floating element can be determined. The relationship between skin friction and the differential capacitance output will also be obtained [9], viz. *τ_w_* = *k_c_*× △*C*. Several sensor prototypes were statically calibrated, each sensor prototype has 10 calibrated values in the 0–100 Pa region, and each calibrated value is tested seven times, as shown in Figure 10. The relationship between skin friction *τ_w_* and the differential capacitance outputs △*C* can be linearly fitted, the fitted results and the calibrated indexes of sensor prototypes are shown in Table 2.

The static calibration results indicate that most sensor prototypes have a fast response, good stability and zero-return; their measurement ranges are 0–100 Pa, their resolution is 0.1 Pa, and the repeatability accuracy and the linearity are both better than 1%. Compared with the preliminary sensor prototypes described in [9], the calibration coefficients (sensitivities) are greatly increased, and the repeatability accuracy and the linearity are also enhanced.

## 5. Validations in a Hypersonic Wind Tunnel

Validation of the sensor prototypes in a hypersonic wind tunnel mainly involves prototype screening and repeatability tests. On these bases, dynamic calibration tests aiming at determining the accuracy of skin friction measurements are explored.

### 5.1. Test Facility and Flow Field Parameters

A flat-plate model is adopted to implement the sensor prototype validation. The length of the flat-plate model is 350 mm, its width is 210 mm, and the material is 45# steel. The model is divided into an upper cover plate and a fuselage. Sensor prototypes are imbedded in the fuselage, and the space between prototypes must be large enough to guarantee that they display no disturbances with each other. The two sensors have a 150 mm space from the sharp leading-edge of the model, and the sensors are symmetrically distributed with a distance of 80 mm along the neutral plane of model. Sensor prototypes and test facility are fixed in the working section of the *Φ* 0.5 m hypersonic wind tunnel of CARDC, as shown in Figure 11a. The Mach number is *M*_∞_ = 6, the angle of attack for the model is α = 0°, and the detailed flow field conditions are presented in Table 3.

3# and 5# sensor prototypes are selected to be validated via screening tests under *P*_0_ = 1.0 MPa conditions. Then, validations in laminar flow and turbulent flow conditions are implemented by changing the Reynolds number of the flow field. The test facility is placed below the nozzle exit and kept far away from the free jet region to reduce the shock effect of hypersonic wind tunnel start-stop. The test facility is rapidly sent into flow field uniform area to implement validation tests when the flow field is stable, and the corresponding Schlieren photo is presented in Figure 11b.

### 5.2. Repeatability Test Results and Analysis

#### 5.2.1. Repeatability Tests under Laminar Flow

The repeatability tests under laminar flow are implemented at *P*_0_ = 0.5 MPa conditions, seven output curves for the 3# and 5# sensor prototypes are separately presented in Figure 12, and their test data are analyzed in Table 4.

The results indicate that the sensor prototypes have the advantage of rapid response, good stability and return-to-zero, and the dynamic pressure values of repeatable tests are in the 7.57–7.87 kPa range. For the 3# sensor prototype, the measured skin friction values are in the 6.84–7.06 Pa range the relevant skin friction coefficients are in the 8.86 × 10^−4^–9.10 × 10^−4^ range with a deviation from the analytic value of 7.25 × 10^−4^ in the 22.21%–25.52% range, and the repeatability accuracy is 1.52%. For the 5# sensor prototype, the measured skin friction values are in the 15.43–15.91 Pa range, the relevant skin friction coefficients are in the 20.06 × 10^−4^–20.86 × 10^−4^ range with a deviation from the analytic value of 7.25 × 10^−4^ in the 176.69%–187.72% range, yet its repeatability accuracy is 2.59%.

#### 5.2.2. Repeatability Tests under Turbulent Flow

Five repeatability tests under turbulent flow and one shock resistance test are separately implemented under *P*_0_ = 2.0 MPa and *P*_0_ = 2.5 MPa conditions. The dynamic pressures of repeated tests under 2.0 MPa conditions are in the 31.97–32.23 kPa range, and six output curves for the 3# and 5# sensor prototypes are separately presented in Figure 13, and the test data are analyzed in Table 5. 

For the 3# sensor prototype, the measured skin friction values are in the 35.52–46.30 Pa range, the relevant skin friction coefficients are in the 11.11 × 10^−4^–14.36 × 10^−4^ range, with deviations from the numerical solution of 112.74 × 10^−4^ under turbulent flow conditions in the 6.99%–12.79% range. The skin friction coefficients gradually increased during the previous five tests. The reason might be that the test-head gradually bulges out of the measured wall due to uncured epoxy resin in the micro-assembly, yet the repeatability accuracy is 2.83% in the last three tests. The measured skin friction value is 54.97 Pa under 2.5 MPa conditions, the related skin friction coefficient is 13.70×10^−4^, and the deviation from the numerical solution (12.26 × 10^−4^) is 11.75%. For the 5# sensor prototype, the measured skin friction values are in the 31.97–32.24 Pa range, the relevant skin friction coefficients are in the 11.45 × 10^−4^–12.12 × 10^−4^ range with deviations in the 4.87%–10.13% range from the numerical solution (12.74 × 10^−4^), and the repeatability accuracy is 3.20%. The measured skin friction value is 42.19 Pa under 2.5 MPa conditions, the related skin friction coefficient is 10.52 × 10^−4^ with a deviation of 14.19% from the numerical solution of 12.26 × 10^-4^.

In total, the results of the repeatability tests indicate that the sensor prototypes possess rapid responses, good stability and return-to-zero, limited by uncured epoxy resin in the micro-assembly, yet the repeatability accuracy is better than 2% under laminar flow conditions and almost 3% under turbulent flow conditions. The deviations between the measured skin friction coefficients and the numerical solutions are almost 10% under turbulent flow conditions, whereas the deviations are large (even more than 100%) under laminar flow conditions, the reason might be the unstudied perceptibility of direct skin friction measurement, which will be discussed next.

### 5.3. Dynamic Calibration Exploration

The 3# and 5# sensor prototypes were also selected to explore dynamic calibration when the Mach number is *M*_∞_ = 6, and the angle of attack for flat-plate model is α = 0°. Dynamic calibrations in laminar flow and turbulent flow conditions are implemented by changing the Reynolds number of the flow field, and the critical unit Reynolds number is about 1.0 × 10^7^.

The varying dynamic pressure method, viz. varying the stable section total pressure of the wind tunnel, is adopted to implement dynamic calibrations under laminar flow and turbulent flow conditions. The total dynamic calibration pressures under laminar flow conditions are in the 0.2–0.6 MPa range; and in the 1.5–2.5 MPa range under turbulent flow conditions. The detailed flow field conditions are presented in Table 6.

#### 5.3.1. Dynamic Calibration in Laminar Flow

Five dynamic calibration tests under laminar flow conditions are implemented under 0.2–0.6 MPa total pressure, and five output curves for the 3# and 5# sensor prototypes are separately presented in Figure 14, and their test data are analyzed in Table 7.

The results indicate the deviations between the measured skin friction coefficients and the analytic values under laminar flow conditions are in the 17.89%–155.22% range for the #3 sensor prototype. Similarly, the deviations between the measured skin friction coefficients and the analytic values are in the 69.91%–162.39% range for the #5 sensor prototype. In brief, the deviations between the measured skin friction coefficients and the analytic values of the two sensor prototypes are almost in the 100%–160% range under laminar flow conditions, and they rapidly increase with increasing total pressure.

#### 5.3.2. Dynamic Calibration under Turbulent Flow

Three dynamic calibration tests under turbulent flow conditions are implemented under 1.5, 2.0 and 2.5 MPa total pressure conditions. Three output curves for the 3# and 5# sensor prototypes are separately presented in Figure 15, and their test data are analyzed in Table 8.

The results indicate the deviations between the measured skin friction coefficients and the numerical solutions are in the 11.75%–23.60% range under turbulent flow conditions for the #3 sensor prototype. Similarly, the deviations between the measured skin friction coefficients and the numerical solutions are in the 0.53%–10.11% range for the #5 sensor prototypes. In brief, the deviations between the measured skin friction coefficients and the numerical solutions of the two sensor prototypes are almost 10% under turbulent flow conditions.

## 6. Discussion

Dynamic calibration results indicate that the deviations between the measured skin friction coefficients and the analytical values under laminar flow conditions are much larger than those seen under turbulent flow conditions. On the one hand, the skin friction coefficients under laminar flow conditions are very small, and equivalent to the sensor prototypes’ system errors induced by perceptibility. On the other hand, the skin friction coefficients under turbulent flow conditions are several times larger than those under laminar flow conditions, and they are much larger than the system errors. Studies indicate that system errors are closely related to many factors, including the flow regime and flow velocity, the shapes and sizes of floating elements, gaps surrounding floating elements, misalignment errors (intrusive or recessive) between floating elements and the measured wall, pressures acting on the lip and surface of floating elements and environmental factors in wind tunnel running. All of these factors are simply analyzed as follows.

### 6.1. Flow State and Velocity

Studies indicate [14], that the height *δ* of the viscous sublayers (*y*+ = 6) for low-speed, low-shear flows and high-speed, turbulent flows is typically in the order of *O* (10^−4^ m) or greater, MEMS sensors have small feature sizes on the order of *O* (10^−6^–10^−5^ m).Thus, the gaps and the protrusions will typically fall within the viscous sublayers, and the flow is considered as being “hydraulically smooth” and the effect on the accuracy of skin friction measurements is negligible. However, the height *δ* of the viscous sublayers for high-speed, high-shear laminar flows (especially for current sensor design) can be on the order of *O* (10^−6^ m) or even smaller, resulting in sensor features not being considered as hydraulically smooth, and this will greatly affect the accuracy of skin friction measurements. Besides, the main relative factors of flow regime and velocity for the flow field of wind tunnel are Reynolds number and Mach number; however, O’Donnell et al. [15] experimentally tested for misalignment effects over Reynolds numbers at Mach 2.67 and Mach numbers at a Reynolds number based on momentum thickness of 10,000, via a parallel-linkage sensor, and no clear dependency could be seen on either the Mach number or Reynolds number.

### 6.2. The Shapes and Sizes of Floating Elements

Different shapes and sizes of floating elements will cause obvious flow disturbances to the flow near the measured wall, and also affect the accuracy of skin friction measurements. Four floating elements with different shapes and sizes are adopted to study the influence rule of the accuracy of skin friction measurements [12], thus floating elements are optimized to minimize their influence on the accuracy. A convex plate of the floating element is designed between its test-head and supporting pole to prevent impurities from entering the moving gaps of unsealed sensing capacitances, as shown in Figure 1a. However, this will cause flow disturbances, and also affect the accuracy of skin friction measurements. Thus, the influence on the accuracy of skin friction measurements induced by floating elements will be studied in depth.

### 6.3. Misalignment Errors

Misalignment errors between the floating elements and the measured wall mainly involve intrusive and recessive areas. Experiments are adopted to study the influence of misalignment errors on the accuracy of skin friction measurements by Meritt et al. [12]. Results indicate that the deviations induced by misalignment errors are not larger than 3% when misalignment errors are not larger than 1.5% of the measured boundary layer thickness, intrusive areas induce negative deviations, and recessive ones induce positive deviations. Furthermore, the deviations can be ignored when misalignment errors are not larger than 1% of the boundary layer thickness. Thus, misalignment errors in the micro-assembly process must be strictly controlled.

### 6.4. Gaps Surrounding Floating Elements

Because the height *δ* of the viscous sublayers for high-speed, high-shear laminar flows are on the order of *O* (10^−6^ m) or even smaller, resulting in the gaps surrounding the floating elements not being considered as hydraulically smooth, the accuracy of skin friction measurements is greatly affected. Under the same intrusive or recessive conditions, experiments are adopted to study the influence rule of the gaps surrounding floating elements on the accuracy [12]. Results indicate that smaller gaps will induce larger deviations, and the deviations will engender sharp deviations (even more than 200%). Besides, similar results of influence are obtained by using numerical simulation methods [16]. Accordingly, the influences of the gaps surrounding the floating elements on the accuracy of skin friction measurements must be studied in depth.

### 6.5. Lip-Force Contributions

Experiments indicate [12], that when the floating element is even with the measured wall, viz. the misalignment error *Z* = 0, the pressure surrounding the floating element is nearly constant without any influence on skin friction measurement. The pressure distribution will engender sharp variations when misalignment errors exist. When the floating element is recessive (*Z*/*δ* = −0.006), the lip pressure is sharp downstream around its trailing edge, and this pressure imbalance will produce lip-force opposing the direction of skin friction, thus reduce the output of the sensor. The influences are opposite for the intrusive (*Z*/*δ* = +0.018) test case and result in a large lip pressure at the leading edge of floating element. In short, recessive areas will bring negative lip-force contribution, and intrusive ones will bring positive lip-force contribution. Furthermore, the lip-force contribution is more significant when the gaps are smaller.

### 6.6. Environmental Factors in Wind Tunnel

The output curves of sensor prototypes have already been analyzed in detail [9]. The results indicate that the influences of environmental factors in a running wind tunnel are tiny, and might be ignored. In total, there are many error sources in skin friction measurements under hypersonic flow, especially under laminar flow conditions. The main factors are listed as follows: the shapes of floating elements, the gaps surrounding the floating elements, misalignment errors and lip-force contributions. Smaller gaps will induce more significant deviations under given misalignment errors. The error sources and their influence rules will be systematically and deeply studied to improve the accuracy of skin friction measurements in the future.

## 7. Conclusions

A novel fabrication method based on visual alignment for MEMS skin friction sensors is presented, and it mainly involves sensor optimization, precise fabrication of key parts, and sensor prototype micro-assembly based on visual alignment. On these bases, static calibrations and validations in a hypersonic wind tunnel are implemented. The main conclusions are:Firstly, the sensor-head is optimized with one pair of differential capacitances; the interface circuit is accordingly optimized, and micro-strip circuit technology is adopted to implement its fabrication with miniaturization and high precision.Secondly, the MEMS skin friction sensor is divided into several parts. Their MEMS technologies and precision machining technologies are studied. The sample parts are fabricated with high precision. Micro-assembly based on visual alignment is adopted to fabricate the sensor prototypes. The precision of fabrication and assembly for sensor prototypes achieve the desired effect.Next, sensor prototypes are statically calibrated via the centrifugal force equivalence method, the measurement ranges are in 0–100 Pa, the resolutions are 0.1 Pa, and the repeatability accuracy and the linearity are better than 1%. In comparison, the calibration coefficients are greatly increased, and the repeatability accuracy and linearity are also improved.Finally, the sensor prototypes are validated in a hypersonic wind tunnel. Screening tests, repeatability tests and dynamic calibrations are implemented. The results indicate that sensor prototypes have the advantages of rapid response, good stability and return-to-zero in hypersonic flow; the repeatability is better than 2% under laminar flow conditions and almost 3% under turbulent flow conditions; the deviations between the measured skin friction coefficients and the numerical solutions are almost 10% under turbulent flow conditions, whereas the deviations are large (even more than 100%) under laminar flow conditions. Besides, the shapes of floating elements, the gaps surrounding floating elements, misalignment errors, lip-force contributions and other error sources of direct skin friction measurements under hypersonic flow are systematically analyzed, and their influence rules will be studied in depth to improve the accuracy of skin friction measurements under laminar flow conditions in the future.

## Figures and Tables

**Figure 1 sensors-19-03803-f001:**
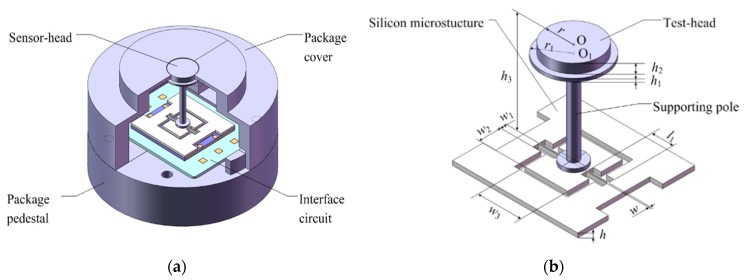
Sensor optimization. (**a**) Overall structure; (**b**) Sensor-head.

**Figure 2 sensors-19-03803-f002:**
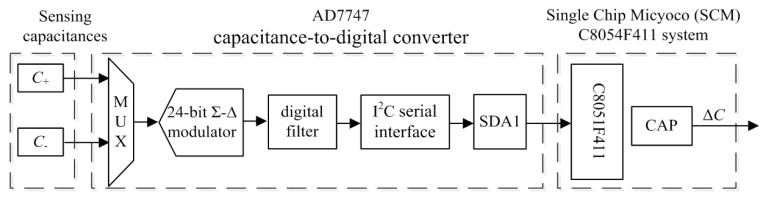
Function module for optimized interface circuit.

**Figure 3 sensors-19-03803-f003:**
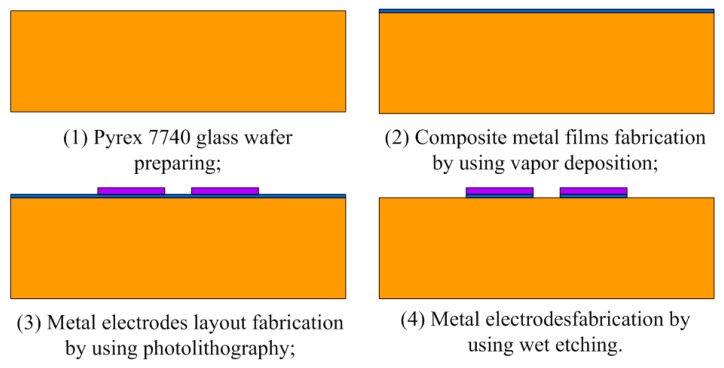
MEMS technological processes for glass-base.

**Figure 4 sensors-19-03803-f004:**
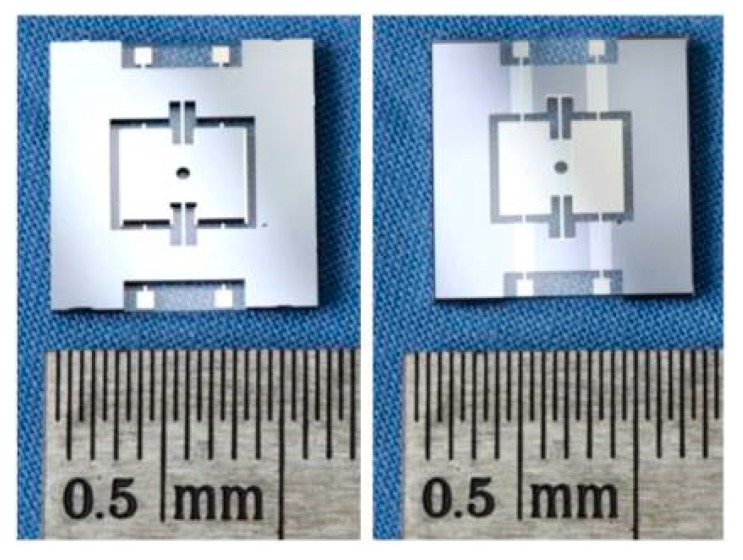
The frontal and rear photos of bonded microstructure.

**Figure 5 sensors-19-03803-f005:**
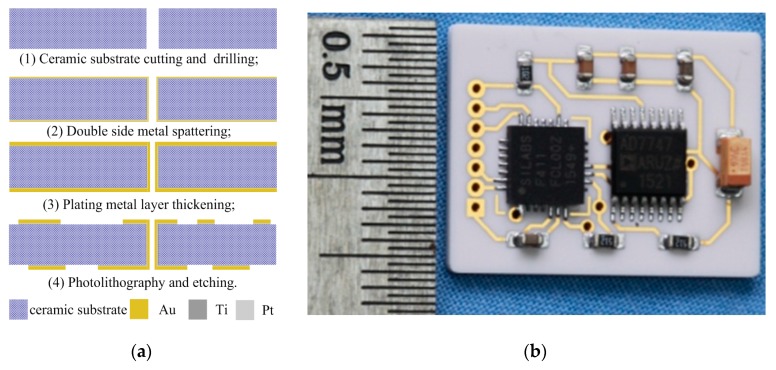
Technological process and photo for the ceramic-based interface circuit. (**a**) Technological process; (**b**) Photo.

**Figure 6 sensors-19-03803-f006:**
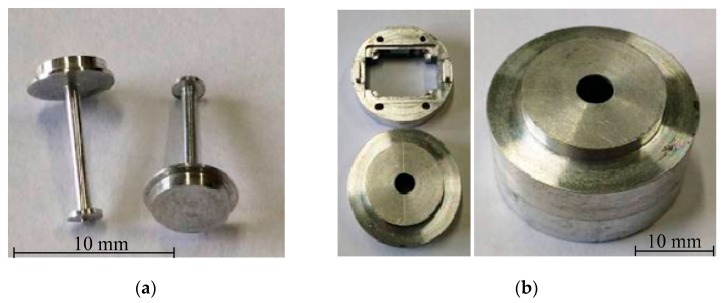
The photos of floating elements and package shells. (**a**) Floating elements; (**b**) Package shells.

**Figure 7 sensors-19-03803-f007:**
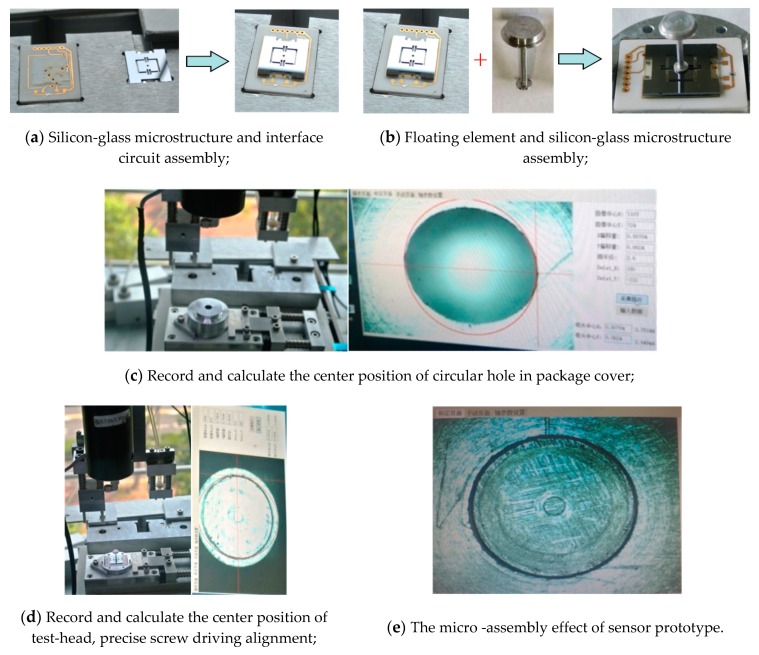
The micro-assembly processes based on visual aligning for MEMS skin friction sensor.

**Figure 8 sensors-19-03803-f008:**
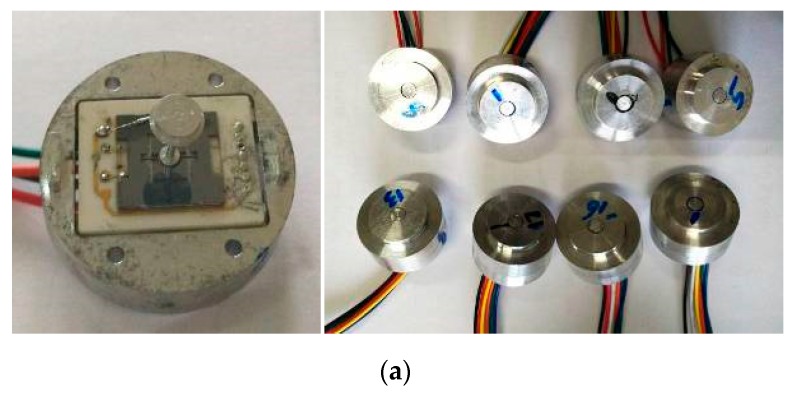
The photos of MEMS skin friction sensor prototypes and package comparison for two-period sensor prototypes. (**a**) Photos of sensor prototypes; (**b**) Prototype based on visual alignment; (**c**) Prototype assembled by hand.

**Figure 9 sensors-19-03803-f009:**
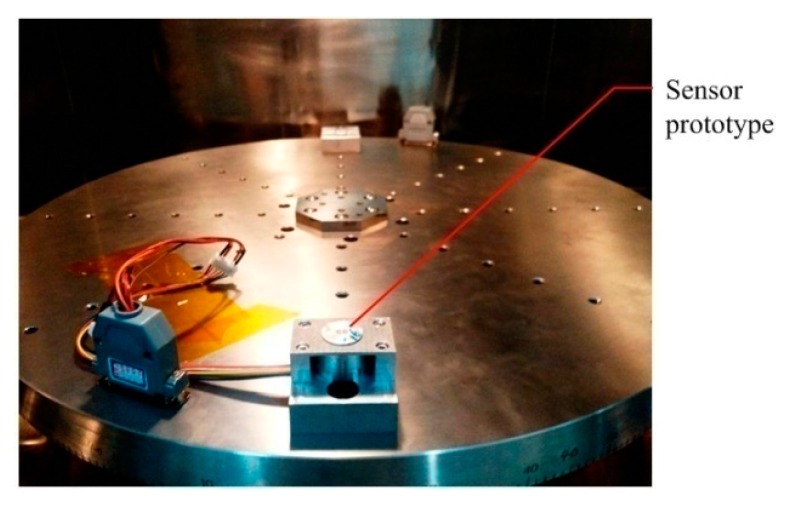
The static calibration device for sensor prototypes.

**Figure 10 sensors-19-03803-f010:**
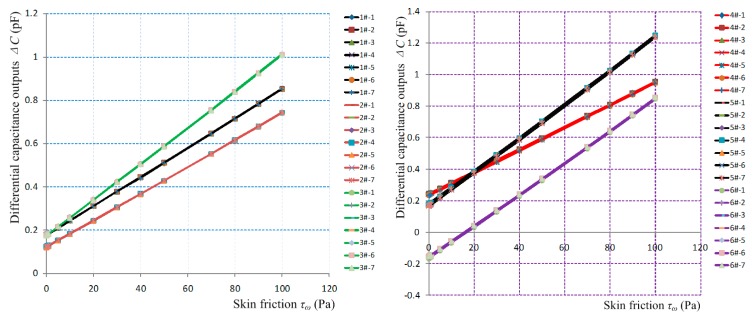
Static calibration curves for sensor prototypes.

**Figure 11 sensors-19-03803-f011:**
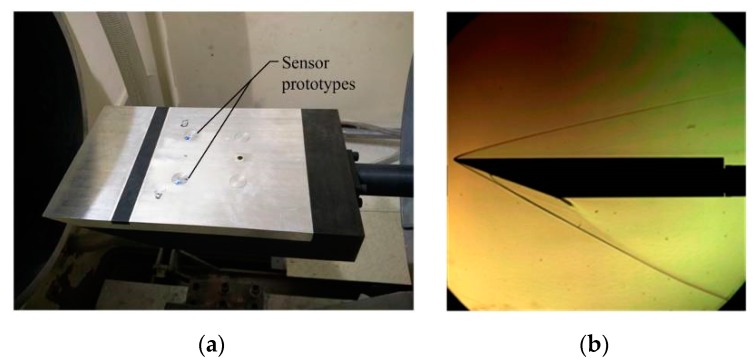
The test facility and Schlieren photo for sensor prototypes. (**a**) The test facility; (**b**) Schlieren photo.

**Figure 12 sensors-19-03803-f012:**
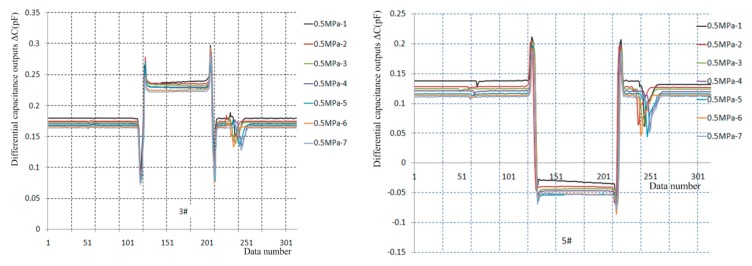
Repeatability test curves under laminar flow conditions.

**Figure 13 sensors-19-03803-f013:**
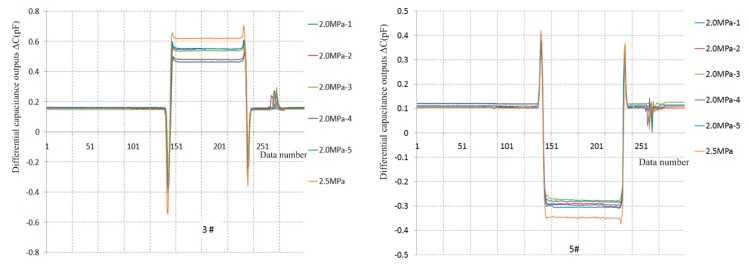
Repeatability test curves in turbulent flow conditions.

**Figure 14 sensors-19-03803-f014:**
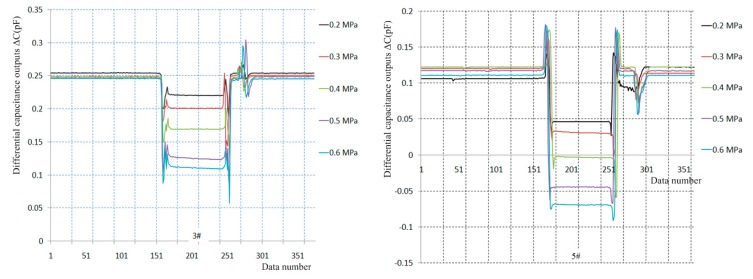
Dynamic calibration curves under laminar flow conditions.

**Figure 15 sensors-19-03803-f015:**
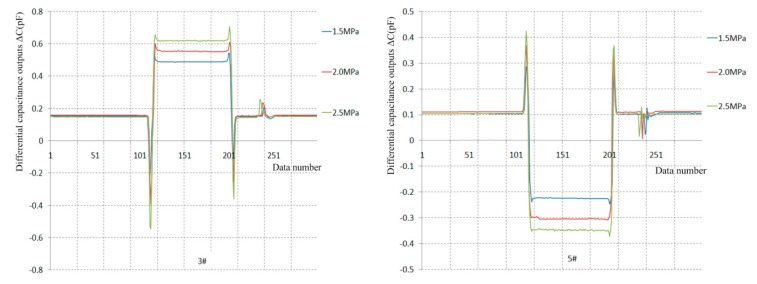
Dynamic calibration curves under turbulent flow conditions.

**Table 1 sensors-19-03803-t001:** The optimized structural parameters for sensor-head.

*w* (μm)	*w*_1_ (μm)	*w*_2_ (μm)	*w*_3_ (μm)	*l*_1_ (μm)	*h*_0_ (μm)	*h* (μm)	*h*_1_ (μm)	*h*_2_ (μm)	*h*_3_ (μm)	*r* (μm)	*r*_1_ (μm)
140	505	2050	4100	2000	10	400	900	300	10,050	2500	3000

**Table 2 sensors-19-03803-t002:** The performance indexes of sensor prototypes static calibration.

Sensor Prototypes	Ranges (Pa)	Resolution (Pa)	Calibration Coefficients (Pa/pF)	Repeatability Accuracy	Linearity
1#	0–100	0.1	148.77	0.157%	0.297%
2#			161.80	0.123%	0.415%
3#			116.67	0.124%	0.417%
4#			140.81	0.634%	0.235%
5#			93.86	0.547%	0.617%
6#			99.67	0.567%	0.996%

**Table 3 sensors-19-03803-t003:** The detailed flow field parameters of the wind tunnel used for sensor validation.

Nominal Mach Number	Total Pressure *P*_0_ (MPa)	Total Temperature *T*_0_ (K)	Static Pressure *P*_∞_ (Pa)	Dynamic Pressure *q*_∞_ (Pa)	Unit Reynolds Number Rel (m^−1^)
6	0.5	417	316.7	7980.4	5.68 × 10^6^
	1.0	437	633.3	1596.1	1.05 × 10^7^
	2.0	458	1266.7	31,921.0	1.94 × 10^7^
	2.5	466	1583.4	39,902.0	2.36 × 10^7^

**Table 4 sensors-19-03803-t004:** The data analysis of repeatability tests under 0.5 Pa total pressure conditions.

Sensor Prototypes	Total Pressure (MPa)	Dynamic Pressure (kPa)	Skin Friction (Pa)	Skin Friction Coefficients	Theoretical Solution	Repeatability Accuracy	Theoretical Deviation ×100%
3#	0.5	7.61	6.84	9.00 × 10^−4^	7.25 × 10^−4^	1.52%	24.14
		7.77	7.07	9.10 × 10^−4^			25.52
		7.87	6.98	8.87 × 10^−4^			22.34
		7.81	6.92	8.86 × 10^−4^			22.21
		7.86	7.06	8.98 × 10^−4^			23.86
		7.66	6.87	8.96 × 10^−4^			23.59
		7.57	6.81	8.99 × 10^−4^			24.00
5#		7.61	15.86	20.86 × 10^−4^	7.25 × 10^−4^	2.59%	187.72
		7.77	15.74	20.26 × 10^−4^			179.45
		7.87	15.78	20.06 × 10^−4^			176.69
		7.81	15.77	20.19 × 10^−4^			178.48
		7.86	15.91	20.25 × 10^−4^			179.31
		7.66	15.55	20.29 × 10^−4^			179.86
		7.57	15.43	20.39 × 10^−4^			181.24

**Table 5 sensors-19-03803-t005:** The data analysis of repeatability tests in 2.0 and 2.5 total pressure conditions.

Sensor Prototypes	Total Pressure (MPa)	Dynamic Pressure (kPa)	Skin Friction (Pa)	Skin Friction Coefficients	Theoretical Solution	Repeatability Accuracy	Theoretical Deviation ×100%
3#	2.0	31.97	35.52	11.11 × 10^−4^	12.74 × 10^−4^	3.18%	12.79
		31.99	37.88	11.84 × 10^−4^			7.06
		32.24	43.95	13.63 × 10^−4^			6.99
		32.21	45.39	14.09 × 10^−4^		2.83%	10.60
		32.23	46.30	14.36 × 10^−4^			12.72
	2.5	40.12	54.97	13.70 × 10^−4^	12.26 × 10^−4^	-	11.75
5#	2.0	31.97	36.62	11.45 × 10^−4^	12.74 × 10^−4^	3.20%	10.13
		31.99	37.16	11.62 × 10^−4^			8.79
		32.24	36.93	11.45 × 10^−4^			10.13
		32.21	39.04	12.12 × 10^−4^			4.87
		32.23	38.96	12.09 × 10^−4^			5.10
	2.5	40.12	42.19	10.52 × 10^−4^	12.26 × 10^−4^	-	14.19

**Table 6 sensors-19-03803-t006:** The detailed flow field parameters of wind tunnel for dynamic calibration exploring.

Nominal Mach Number	Total Pressure (MPa)	Total Temperature (K)	Static Pressure (Pa)	Dynamic Pressure (Pa)	Unit Reynolds Number (m^−1^)
6	0.2	393	126.7	3192	2.51 × 10^6^
	0.3	403	190.1	4788	3.61 × 10^6^
	0.4	411	228.0	6384	4.67 × 10^6^
	0.5	416	316.6	7980	5.70 × 10^6^
	0.6	422	380.0	9576	6.69 × 10^6^
	1.5	449	950.0	23,941	1.51 × 10^7^
	2.0	458	1266.7	31,921	1.94 × 10^7^
	2.5	466	1583.4	39,902	2.36 × 10^7^

**Table 7 sensors-19-03803-t007:** The data analysis of dynamic calibration under laminar flow conditions.

Sensor Prototypes	Total Pressure (MPa)	Dynamic Pressure (kPa)	Skin Friction (Pa)	Skin friction Coefficients	Theoretical Solution	Theoretical Deviation ×100%
3#	0.2	3.95	3.07	12.85 × 10^−4^	10.9 × 10^−4^	17.89
	0.3	5.61	3.91	14.32 × 10^−4^	9.1 × 10^−4^	57.36
	0.4	9.26	6.01	15.41 × 10^−4^	8.0 × 10^−4^	92.63
	0.5	14.19	8.66	16.38 × 10^−4^	7.2 × 10^−4^	127.50
	0.6	15.75	9.21	17.10 × 10^−4^	6.7 × 10^−4^	155.22
5#	0.2	5.63	3.04	18.52 × 10^−4^	10.9 × 10^−4^	69.91
	0.3	8.36	4.31	19.41 × 10^−4^	9.1 × 10^−4^	113.30
	0.4	11.78	6.17	19.10 × 10^−4^	8.0 × 10^−4^	138.75
	0.5	15.12	8.27	18.35 × 10^−4^	7.2 × 10^−4^	154.86
	0.6	16.90	9.61	17.58 × 10^−4^	6.7 × 10^−4^	162.39

**Table 8 sensors-19-03803-t008:** The data analysis of dynamic calibration under turbulent flow conditions.

Sensor Prototypes	Total Pressure (MPa)	Dynamic Pressure (kPa)	Skin Friction (Pa)	Skin Friction Coefficients	Numerical Solution	Theoretical Deviation ×100%
3#	1.5	39.15	23.96	16.34 × 10^−4^	13.22 × 10^−4^	23.60
	2.0	46.22	32.24	14.34 × 10^−4^	12.74 × 10^−4^	12.56
	2.5	54.97	40.12	13.70 × 10^−4^	12.26 × 10^−4^	11.75
5#	1.5	31.50	23.96	13.15 × 10^−4^	13.22 × 10^−4^	0.53
	2.0	37.18	32.24	11.53 × 10^−4^	12.74 × 10^−4^	9.50
	2.5	44.22	40.12	11.02 × 10^−4^	12.26 × 10^−4^	10.11

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
