# Peer review of "Fabrication and Hypersonic Wind Tunnel Validation of a MEMS Skin Friction Sensor Based on Visual Alignment Technology"

_sensors, 2019, doi:10.3390/s19173803_

Round 1
Reviewer 1 Report
Please see attached.

Author Response
Dear professor:
Thank you very much for your pertinent comments on our paper.
We seriously revise it one by one, please see the attachment.
We trouble you for your guidance and help again.
The authors
The authors’ responses are given as follows:
Both the writing style and English grammar should be improved. Sentences are often very hard to follow. For instance, abstract: “...and its distribution, own small volunteer, higher reliability...” volunteer? And: “...also very important to skin friction measuring research in hypersonic wind tunnel.” Basically, every other sentence comprises style and grammar mistakes.
Response 1: The language of our paper has already been re-edited basing on reviewers’ comments, as shown in manuscript-revision submitted.
Static calibration and Figure 12: it is unclear what ∆C is (and the dimension) and pF. Graphs should be improved and it may not be necessary to label all the points of each curve in the legend (cannot really distinguish the points anyway).
Response 2: ∆C represents differential capacitance outputs of the sensors, and is also given in Figure 12, as shown in manuscript-revision; pF is the capacitance unit, viz. one trillionth of a Farad. Besides, the points of each curve in the legend only represent the calibrated skin friction values, each sensor prototype owns 10 calibrated values in 0-100 Pa areas, and each calibrated value is tested for seven times; thus you need not to distinguish the points anyway, and this also means the sensors own better linearity and repeatability in static calibrations.
In the wind tunnel experiment, temperature must be wildly varying (much colder when the flow is on). How does this affect the output of the sensors? Can this be incorporated in the static calibration experiment when the static calibration experiment is placed in a temperature controlled chamber? Would this not be the largest source of error for the ’theoretical deviation’? This issue seems neglected in Section 6.
Response 3:
In these wind tunnel experiments, the total temperature of stable section is heated as 450 K (at Ma=6) for air anti-condensation in hypersonic wind tunnel. The temperature on the surface of model measured is the recover temperature of the flow, and the recover temperature is always about 40℃-50℃ in actual test. Furthermore, the signal output microstructure is isolated from the hypersonic flow field by the supporting pole, and also the time of skin friction experiment is less than 30 seconds, and temperature will not affect the output of the sensors. Besides, it can be seen from Fig13-16 in our manuscript that zero-shift of sensor prototypes varies tinily in one test, which might be induced by airstream disturbance and pressure variation in working section.
We think that the temperature will not affect the performance of the sensors, and is not considered in static calibration experiments. This issue is only mentioned as “The output curves for sensor prototype has already been analyzed in details [9]. Results indicate, the influences of environmental factors in wind tunnel running is tiny, and they might be ignored” in Section 6. Thus, the influence of temperature might be ignored in wind tunnel running and static calibration.

Reviewer 2 Report
In this manuscript, the authors developed a skin friction sensor prototype with a fast response and better stability. However, from my understanding, the authors are not able to provide enough information to introduce the novelty, mechanism, the working process and conclusions. Therefore, the authors need to clarify the concerns.
Here, the authors have not presented the novelty of their prototype. I can not figure out the difference between the paper and the following paper:Wang, X.; Zhu,T.; Xu, X. B.; et al. Fabrication, calibration and proof experiments in a hypersonic wind tunnel for a novel MEMS skin friction sensor [J]. Microsystem Technologies, 2017, 23(8), 3601-3611
From my perspective, the authors seem to have the same design, but slightly changed the parameters and maybe introduce the visual aligning technology. Many figures are quite similar.
The working mechanism not well described. For the readers, they may not easy to catch the principles of the devices. The authors failed to provide this information. There are a lot of grammar issues. They need to pay more attention to their English writing. The working process is not well described. No figure and video are provided to show how this works.Author Response
Dear professor:
Thank you very much for your pertinent comments on our paper.
We seriously revise it one by one, please see the attachment.
We trouble you for your guidance and help again.
The authors
The authors’ responses are given as follows:
In this manuscript, the authors developed a skin friction sensor prototype with a fast response and better stability. However, from my understanding, the authors are not able to provide enough information to introduce the novelty, mechanism, the working process and conclusions. Therefore, the authors need to clarify the concerns.
Here, the authors have not presented the novelty of their prototype. I cannot figure out the difference between the paper and the following paper: Wang, X.; Zhu,T.; Xu, X. B.; et al. Fabrication, calibration and proof experiments in a hypersonic wind tunnel for a novel MEMS skin friction sensor [J]. Microsystem Technologies, 2017, 23(8), 3601-3611. From my perspective, the authors seem to have the same design, but slightly changed the parameters and maybe introduce the visual aligning technology. Many figures are quite similar.
Response 1:
In fact, the paper published (“Fabrication, calibration and proof experiments in a hypersonic wind tunnel for a novel MEMS skin friction sensor [J]. Microsystem Technologies, 2017, 23(8), 3601-3611”) is our preliminary work on the same MEMS skin friction sensor, and has been simply introduced in the 3rd paragraph of part 1 - introduction in this paper. However, limited by fabrication errors of sensor prototypes in preliminary work, test accuracy cannot be estimated.
In this paper, aiming at the problem of skin friction test accuracy induced by existing sensor fabrication and assembly errors, a novel fabrication technology based on visual aligning is presented. Sensor optimization, prototypes fabrication, static calibrations and validation in hypersonic wind tunnel are presented. Besides, dynamic calibrations in laminar and turbulent flow conditions are implemented, and the error resources of skin friction measurements and their influence rules are systematically discussed and analyzed to improve test accuracy in laminar flow conditions. Totally, fabrication technologies based on visual aligning basically resolve existing sensor fabrication errors, and is also very important to skin friction measurement in hypersonic wind tunnel.
We think that the difference between the two papers and the novelty of this paper have been clarified as mentioned above, and the reviewer’s perspective that “the authors seem to have the same design, but slightly changed the parameters and maybe introduce the visual aligning technology, many figures are quite similar” has also been clarified.
The working mechanism not well described. For the readers, they may not easy to catch the principles of the devices. The authors failed to provide this information. The working process is not well described. No figure and video are provided to show how this works.
Response 2:
Since the paper published is our preliminary work on the same MEMS skin friction sensor, and has already been simply introduced (“A novel MEMS skin friction sensor was developed by us [8,9], which adopted that floating element even with the measured wall and signal output microstructure being isolated from hypersonic flow field. Its test-head is connected to a clamped-clamped beam with sensing capacitances through a supporting pole, thus skin friction sensed by test-head is transferred to the clamped-clamped beam and drive bilateral vibratory electrodes of sensing capacitances to generate torsional angle displacement; then, the measured skin friction can be solved by differentially calculating the variations of sensing capacitances”) in the 3rd paragraph of part 1 introduction in this paper, “the working mechanism” is not well described to avoid repetition. If the readers want to catch the principles of the sensors, we think they can catch this information from the published paper [9].
There are a lot of grammar issues. They need to pay more attention to their English writing.
Response 3: The language of our paper has already been re-edited basing on reviewers’ comments, as shown in manuscript-revision submitted.

Reviewer 3 Report
This paper aims to develop a MEMS skin friction sensor. The main issue with the paper is that the results have not been presenting well.
Also, the quality of the images needs to improve. Authors need to do a better job of putting the paper together and explaining their work and contribution. Below are some comments and suggestions.
1) Perhaps the weakest point of the paper is its introduction section. Very few papers have been cited and most of them are outdated. Authors need to expand the introduction by discussion recent MEMS sensors. Below papers might be helpful.
I) "MEMS piezoresistive flow sensors for sleep apnea therapy," Sensors and Actuators A: Physical, vol. 279, pp. 577-585, 2018.
II) "Design and applications of MEMS flow sensors: A review," Sensors and Actuators A: Physical, 2019.
III) A. Hald, H. Marquardt, P. Herzogenrath, J. Scheible, J. Lienig, and J. Seelhorst, "Full Custom MEMS Design: 2.5D Fabrication-Process Simulation for 3D Field-Solver-Based Circuit Extraction," Ieee Sensors Journal, vol. 19, pp. 5710-5717, Jul 2019.
IV) G. Imamura, K. Shiba, G. Yoshikawa, and T. Washio, "Free-hand gas identification based on transfer function ratios without gas flow control," Scientific Reports, vol. 9, Jul 2019.
V) E. G. Keeler, C. Zou, and L. Y. Lin, "Optically Accessible MEMS Resonant Mass Sensor for Biological Applications," Journal of Microelectromechanical Systems, vol. 28, pp. 494-503, Jun 2019.
VI) C. C. Nguyen, V. K. T. Ngo, H. Q. Le, and W. L. Li, "Influences of relative humidity on the quality factors of MEMS cantilever resonators in gas rarefaction," Microsystem Technologies-Micro-and Nanosystems-Information Storage and Processing Systems, vol. 25, pp. 2767-2782, Jul 2019.
VII) J. H. Xu, K. T. C. Chai, G. Q. Wu, B. B. Han, E. L. C. Wai, W. Li, et al., "Low-Cost, Tiny-Sized MEMS Hydrophone Sensor for Water Pipeline Leak Detection," Ieee Transactions on Industrial Electronics, vol. 66, pp. 6374-6382, Aug 2019.
2) Please add scale bar for image 6.
3) Figure 7 needs attention. Please remove the text from the image and add it in the figure caption. Also, try to use a sharper image with higher quality.
4) You may combine figure 8, 9 and 10.
5) Figure 11 needs to have clear X axis and Y axis description. Just having the unit is not sufficient.
6) Figure 11 and 13 are very confusing. Authors need to use a different way to visualize to present the results of the prototype. Perhaps use the error bar to present the results.
7) A clear description and explanation of figure 11 are required. Just presenting the results is not sufficient. Authors need to discuss the results and compare the calibration results by previous papers and comment on the performance of their sensor.
8) what is the value of the sensitivity for the sensor and how does it stand among previously reported sensors?
9) Figure 14 needs to be replaced with a sharper image. Please be consistent with the fonts and size. It is very difficult to read the numbers in this figure. Also, ensure to have a description for X and Y axis. (Same comments goes for Figure 15).
10) it is not clear how the experimental setup for laminar and turbulent flow are fabricated. An image or schematic for each case would help to make it clear for the readers.
11) Why the repeatability of results reduced in laminar flow (figure 15)?
12) what is different between the results presented in figure 15 and 16?
13)Authors need to clearly state how this work is different by their previous publications “Fabrication, calibration and proof experiments in hypersonic wind tunnel for a novel MEMS skin friction sensor, Microsyst Technol (2017) 23:3601–3611.
Author Response
Dear professor:
Thank you very much for your pertinent comments on our paper.
We seriously revise it one by one, please see the attachment.
We trouble you for your guidance and help again.
The authors
The authors’ responses are given as follows:
1) Perhaps the weakest point of the paper is its introduction section. Very few papers have been cited and most of them are outdated. Authors need to expand the introduction by discussion recent MEMS sensors. Below papers might be helpful.
Response 1:
Direct skin friction measurement is a relatively smaller and more difficult field for study, especially in hypersonic flow, and little progresses has been made in recent years. This is the reason why very few papers have been cited and most of them are outdated. Direct skin friction measurements used in hypersonic wind tunnel mainly involves various conventional skin friction balances [1-5], and the authors have not found any other MEMS skin friction sensors used in hypersonic wind tunnel yet. The papers reviewer provided mainly involve “respiratory system response” , “flow velocity or flow measurements in low-speed flow”, “customized electrostatic analysis flow for MEMS devices” , “gas identification protocol”, “biological applications”, “the effects of relative humidity on the Q factor of MEMS resonators”, “water pipeline leak detection” and so on, but they don’t involve direct skin friction measurements in hypersonic flow.
The authors have made great efforts to seek new papers about direct skin friction measurements in hypersonic flow from internet, but we have not got any new papers published in recent years.
I) "MEMS piezoresistive flow sensors for sleep apnea therapy," Sensors and Actuators A: Physical, vol. 279, pp. 577-585, 2018.
This paper presents a MEMS liquid crystal polymer (LCP) used in the membrane-based pressure sensor, has been found highly useful as a flow sensor, and the LCP MEMS flow sensor is able to detect respiratory system response from inhalation and exhalation.
II) "Design and applications of MEMS flow sensors: A review," Sensors and Actuators A: Physical, 2019.
This paper presents an overview of the work done on design and development of MEMS-based flow sensors in recent years, and these MEMS flow sensors can be mainly used in flow velocity or flow measurements in low-speed flow.
III) A. Hald, H. Marquardt, P. Herzogenrath, J. Scheible, J. Lienig, and J. Seelhorst, "Full Custom MEMS Design: 2.5D Fabrication-Process Simulation for 3D Field-Solver-Based Circuit Extraction," Ieee Sensors Journal, vol. 19, pp. 5710-5717, Jul 2019.
This paper presents a customized electrostatic analysis flow for MEMS devices, features a 2.5D fabrication-process simulation.
IV) G. Imamura, K. Shiba, G. Yoshikawa, and T. Washio, "Free-hand gas identification based on transfer function ratios without gas flow control," Scientific Reports, vol. 9, Jul 2019.
This paper presents a novel gas identification protocol based on a transfer function ratio (TFR) that is intrinsically independent of a gas input pattern.
V) E. G. Keeler, C. Zou, and L. Y. Lin, "Optically Accessible MEMS Resonant Mass Sensor for Biological Applications," Journal of Microelectromechanical Systems, vol. 28, pp. 494-503, Jun 2019.
This paper presents an optically accessible MEMS resonant mass sensor; it is only used for biological applications.
VI) C. C. Nguyen, V. K. T. Ngo, H. Q. Le, and W. L. Li, "Influences of relative humidity on the quality factors of MEMS cantilever resonators in gas rarefaction," Microsystem Technologies-Micro-and Nanosystems-Information Storage and Processing Systems, vol. 25, pp. 2767-2782, Jul 2019.
This paper presents the effects of relative humidity on the Q factor of MEMS resonators in wide range of gas rarefaction conditions.
VII) J. H. Xu, K. T. C. Chai, G. Q. Wu, B. B. Han, E. L. C. Wai, W. Li, et al., "Low-Cost, Tiny-Sized MEMS Hydrophone Sensor for Water Pipeline Leak Detection," Ieee Transactions on Industrial Electronics, vol. 66, pp. 6374-6382, Aug 2019.
This paper presents MEMS hydrophone sensor, and it is only used for water pipeline leak detection.
2) Please add scale bar for image 6.
Response 2: The scale bar for image 6 has been added, as shown in manuscript-revision submitted.
3) Figure 7 needs attention. Please remove the text from the image and add it in the figure caption. Also, try to use a sharper image with higher quality.
Response 3: The text from image 7 has been removed and added in the figure caption,and the image has already been edited again, as shown in manuscript-revision submitted.
4) You may combine figure 8, 9 and 10.
Response 4: Figure 8 and figure 9 are combined as new figure 8; figure 10 is the static calibration device, we think it is better to be separated; the figure numbers are accordingly revised, as shown in manuscript-revision submitted.
5) Figure 11 needs to have clear X axis and Y axis description. Just having the unit is not sufficient.
Response 5: The X axis and Y axis descriptions of this figure are added, as shown in manuscript-revision submitted.
6) Figure 11 and 13 are very confusing. Authors need to use a different way to visualize to present the results of the prototype. Perhaps use the error bar to present the results.
Response 6: Each curve in Figure 11 includes several values of skin friction - differential capacitance output relationship in static calibrations, and each value represents one original test curve, there are many original test curves in static calibrations. Thus, the curves are trimmed as shown in Figure 11, and the curves also present better linearity and repeatability in static calibrations. Each curve in Figure 13 includes only one value of skin friction - differential capacitance output relationship in wind tunnel tests, the seven curves present the whole skin friction measurements and the repeatability of seven wind tunnel tests in the same P0=0.5 MPa conditions. Accordingly, we think that Figure 11 and 13 are not very confusing.
7) A clear description and explanation of figure 11 are required. Just presenting the results is not sufficient. Authors need to discuss the results and compare the calibration results by previous papers and comment on the performance of their sensor.
Response 7: The clear description of figure 11 is revised as: “Several sensor prototypes are statically calibrated, each sensor prototype owns 10 calibrated values in 0 to 100 Pa areas, and each calibrated value is tested for seven times, as shown in Figure 10”. The calibration results are compared with previous papers is added as: “Compared with the preliminary sensor prototypes [9], the calibration coefficients (sensitivity) are largely increased, and the repeatability accuracy and linearity are also promoted”.
8) what is the value of the sensitivity for the sensor and how does it stand among previously reported sensors?
Response 8: The value of the sensitivity is the calibration coefficients in table 2, and the calibration coefficients are multiplied by the differential capacitance outputs to achieve skin friction measured. Compared with the preliminary sensor prototypes [9], the calibration coefficients (sensitivity) are largely increased by increasing capacitance electrodes and reducing the initial capacitance gap between their vibratory electrode and fixed electrode.
9) Figure 14 needs to be replaced with a sharper image. Please be consistent with the fonts and size. It is very difficult to read the numbers in this figure. Also, ensure to have a description for X and Y axis. (Same comments goes for Figure 15).
Response 9: Images in figure 14 -16 have been edited again, descriptions for X and Y axis have been added, as shown in manuscript-revision submitted.
10) it is not clear how the experimental setup for laminar and turbulent flow are fabricated. An image or schematic for each case would help to make it clear for the readers.
Response 10: The experimental setup for laminar and turbulent flow is implemented by changing the Reynolds number of flow field, and the critical unit Reynolds number is about 1.0×107, if the unit Reynolds number is bigger than the critical unit Reynolds number, it is in turbulent flow conditions; if the unit Reynolds number is smaller than the critical unit Reynolds number, it is in laminar flow conditions. The setup of unit Reynolds number can be seen in table 6 for the readers. The related descriptions is revised as : “Dynamic calibrations in laminar flow and turbulent flow conditions are implemented by changing the Reynolds number of flow field, and the critical unit Reynolds number is about 1.0×107. Varying dynamic pressure method, viz. varying stable section total pressure of wind tunnel, is adopted to implement dynamic calibrations in laminar flow and turbulent flow conditions”, as shown in the 1st paragraph of part 5.3 in manuscript-revision submitted.
11) Why the repeatability of results reduced in laminar flow (figure 15)?
Response 11: Figure 15 presents the deviations between measured skin friction coefficients and analytic value in 0.2 MPa to 0.6 MPa total pressure (laminar flow) conditions to check skin friction test accuracy; there exist no repeatability of results. The results indicate deviations in laminar flow conditions are almost in 100% to 160% area, the error resources of skin friction measurements and their influence rules are systematically discussed and analyzed to improve test accuracy in laminar flow conditions in part 6.
12) what is different between the results presented in figure 15 and 16?
Response 12: Figure 15 presents the deviations between measured skin friction coefficients and analytic value in 0.2 MPa to 0.6 MPa total pressure (laminar flow) conditions; Figure 16 presents the deviations between measured skin friction coefficients and analytic value in 1.5 MPa, 2.0 MPa and 2.5 MPa total pressure (turbulent flow) conditions. They are the different parts of dynamic calibrations.
13)Authors need to clearly state how this work is different by their previous publications “Fabrication, calibration and proof experiments in hypersonic wind tunnel for a novel MEMS skin friction sensor, Microsyst Technol (2017) 23:3601–3611.
Response 13:
In fact, the previous publications (“Fabrication, calibration and proof experiments in hypersonic wind tunnel for a novel MEMS skin friction sensor, Microsyst Technol (2017) 23:3601–3611”) is our preliminary work on the same MEMS skin friction sensor, and has been simply introduced in the 3rd paragraph of part 1 introduction in this paper. However, limited by imperfect fabrication precision of assembly and package in our preliminary work, test accuracy cannot be estimated.
In this paper, aiming at the problem of skin friction test accuracy induced by existing sensor fabrication and assembly errors, a novel fabrication technology based on visual aligning is presented. The optimization for existing sensor-head, prototypes fabrication, static calibrations and validation in hypersonic wind tunnel are presented. Besides, dynamic calibrations in laminar and turbulent flow conditions are implemented, and the error resources of skin friction measurements and their influence rules are systematically discussed and analyzed to improve test accuracy in laminar flow conditions next. Totally, fabrication technologies based on visual aligning basically resolve existing sensor fabrication and assembly errors, and is also very important to skin friction measurement in hypersonic wind tunnel. We think that the difference between the two papers and the novelty of this paper have been clarified as mentioned above.

Round 2
Reviewer 1 Report
I believe the authors adressed my comments well.
Author Response
Dear professor:
Thank you very much for your pertinent comments on our paper again.
We try our best to improve the quality of English, as shown in manuscript-revision.
We trouble you for your guidance and help.
The authors
Reviewer 2 Report
Although the authors figure out the difference, they should also describe the basic principle in the paper to help the readers to understand and lead them to read the previous paper, and carefully describe what has been corrected or improved. I am not satisfied with the presenting style, it is more like an experimental report, rather than a paper for people from different areas to read and study. They also need to figure out what the data means in each figures. And how good they are comparing with the previous work or other related work.
There are many grammar errors. The authors should improve the quality much harder. For example, "sensor prototypes has".......
The graphic design is bad. Figure 7 should have outlines to separate each procedure. Figure 8 also needs to be improved.
Author Response
Dear professor:
Thank you very much for your pertinent comments on our paper again.
We seriously revise it, please see the attachment.
We trouble you for your guidance and help.
The authors
The authors’ responses are given as follows:
Although the authors figure out the difference, they should also describe the basic principle in the paper to help the readers to understand and lead them to read the previous paper, and carefully describe what has been corrected or improved. I am not satisfied with the presenting style, it is more like an experimental report, rather than a paper for people from different areas to read and study.
Response:
The basic principle of the sensor has been involved in the 3rd paragraph of part 1 - introduction in this paper, and the related contents are revised as “A novel MEMS skin friction sensor was developed by us [8, 9], which adopted the principle that floating element even with the measured wall and signal output microstructure being isolated from hypersonic flow field. Its test-head was connected to a clamped-clamped beam with sensing capacitances through a supporting pole. Thus, skin friction sensed by test-head was transferred to the clamped-clamped beam and drove bilateral vibratory electrodes of sensing capacitances to generate torsional angle displacement. Then, the measured skin friction could be solved by differentially calculating the variations of sensing capacitances.”
The contents have been corrected and improved have been involved in the last paragraph of part 1 - introduction in this paper, and the related contents are revised as “Aiming at the demerit of the test accuracy induced by fabrication errors, a novel fabrication scheme based on visual aligning is proposed, and it mainly involves the optimizations of sensor-head and interface circuit, precise fabrications, micro-assembly based on visual aligning, prototypes fabrication, static calibration, validation and dynamic calibrations exploring in hypersonic wind tunnel.”
They also need to figure out what the data means in each figures.
Response:
The data means in each figure are described via its X axis and Y axis descriptions, and they are also revealed by the related data analysis tables and the summarized descriptions in detail.
And how good they are comparing with the previous work or other related work.
Response:
Compared with the previous work, the improvements have been involved in the last paragraph of part 3 Sensor Fabrications Based on Visual Aligning, viz. “The precision of fabrication and micro-assembly for sensor prototypes based on visual aligning achieves desired package effect”; in the last paragraph of part 4 static calibrations, viz. “Compared with the preliminary sensor prototypes [9], the calibration coefficients (sensitivities) are largely increased, and the repeatability accuracy and the linearity are also promoted”; and in the last paragraph of part 5.2 Repeatability tests, viz. “The deviations between the measured skin friction coefficients and the numerical solutions are almost 10% in turbulent flow conditions; whereas the deviations are large (even more than 100%) under laminar flow conditions”. Besides, the dynamic calibrations are explored in part 5.3 Dynamic calibrations, and the error resources of direct skin friction measurements in hypersonic flow are systematically analyzed in part 6 discussions.
There are many grammar errors. The authors should improve the quality much harder. For example, "sensor prototypes has".......
Response:
We try our best to improve the quality of English, as shown in manuscript-revision.
The graphic design is bad. Figure 7 should have outlines to separate each procedure. Figure 8 also needs to be improved.
Response:
Figure 7 and Figure 8 are improved, as shown in manuscript-revision.
Reviewer 3 Report
Authors have applied all the comments. I am comfortable with accepting this paper in its present format.
Author Response

(The authors gave the same response as above.)
